Fecal microbiome analysis in patients with metabolic syndrome and type 2 diabetes

Sinisterra Loaiza Laura Isabel 1
http://orcid.org/0000-0001-6906-3374 Fernández-Edreira Diego 2
Liñares-Blanco Jose 2
Cepeda Alberto 1
Cardelle-Cobas Alejandra 1
http://orcid.org/0000-0003-0413-5677 Fernandez-Lozano Carlos 2 carlos.fernandez@udc.es
1 Department of Analytical Chemistry, Nutrition and Bromatology, Faculty of Veterinary Medicine, Universidade de Santiago de Compostela, Campus de Lugo , Lugo , Spain
2 Machine Learning in Life Sciences Laboratory, Department of Computer Science and Information Technologies, Universidade da Coruña (CITIC) , A Coruña , Spain
Khan Haseeb
Electronic publication date: 2025 Jun 11
Publication date: 2025
Volume: 13
Electronic Location ID: e19108
Received 2024 Sep 4; Accepted 2025 May 13
Copyright: © 2025 Sinisterra Loaiza et al.
Copyright year: 2025
Copyright holder: Sinisterra Loaiza et al.
License: This is an open access article distributed under the terms of the Creative Commons Attribution License, which permits unrestricted use, distribution, reproduction and adaptation in any medium and for any purpose provided that it is properly attributed. For attribution, the original author(s), title, publication source (PeerJ) and either DOI or URL of the article must be cited.
License URL: https://creativecommons.org/licenses/by/4.0/

Keywords: Bioinformatics, Microbiome, Type 2 diabetes, Metabolic syndrome, Biomarkers

Funding: CyTED, Spain and National Organism for Science and Technology, IBEROBDIA P918PTE0409 Spanish Ministry of Economy and Competitiveness PCI2018-093245, PCI2018-093284 Ministry of Culture, Education, Vocational Training, and Universities and the Galician Universities University System of Galicia (CIGUS) CESGA (Centro de Supercomputación de Galicia) Spanish Ministry of Universities by means of the Margarita Salas RSUC.UDC.MS06 This work was supported by the CyTED, Spain and National Organism for Science and Technology, founding the IBEROBDIA project (P918PTE0409). This work was financially supported by the Spanish Ministry of Economy and Competitiveness through the State Program of I+D+I Oriented to the Challenges of Society 2017–2020 (International Joint Programming 2018), projects (PCI2018-093245, and PCI2018-093284). CITIC is funded by the Xunta de Galicia through the collaboration agreement between the Ministry of Culture, Education, Vocational Training, and Universities and the Galician universities for the strengthening of research centers in the University System of Galicia (CIGUS). This work was supported by the CESGA (Centro de Supercomputación de Galicia), providing computing resources and related technical support that contributed to the research results reported in this article. Jose Liñares-Blanco’s work was financed by the Spanish Ministry of Universities by means of the Margarita Salas (RSUC.UDC.MS06) linked to the European Union through the NextGenerationEU program. The funders had no role in study design, data collection and analysis, decision to publish, or preparation of the manuscript.

==============================
Background

Metabolic syndrome (MS) and type 2 diabetes (T2D) are metabolically related diseases with rising global prevalence and increasingly evident links to the intestinal microbiota. Research suggests that imbalances in microbiota composition may play a crucial role in their pathogenesis. Specific population cohorts, such as the one in Galicia, Spain, offer the opportunity to analyze microbiota patterns within a distinct geographical and genetic context. This study was performed to investigate the relationship between the intestinal microbiota and MS and T2D.

Methods

A cohort of 79 volunteers was analyzed over a 2-year study period. Recruitment posed significant challenges because of strict inclusion criteria (918PTE0540; PCI2018-093284), which required participants to be free from chronic medications and have a moderate to high risk of developing T2D. Volunteers were classified based on their serum glucose levels, body mass index, and the presence or absence of MS. To analyze the microbiota composition, amplicon sequencing of 16S rRNA genes was performed on stool samples. Alpha diversity was assessed using the Chao and Shannon indices, while beta diversity was evaluated using permutational analysis of variance with Bray–Curtis and Chao distances. Differential abundance analysis was conducted using the LinDA method.

Results

In patients with MS, we observed a higher Firmicutes/Bacteroidetes ratio and an increased prevalence of Blautia compared to healthy patients. than in healthy individuals. Other enriched taxa in patients with MS included Tyzerella, Streptococcus, and Ruminococcus callidus. In patients with T2D, we observed a higher Bacteroidetes/Firmicutes ratio and a decrease in the phylum Actinobacteria compared with healthy individuals. Taxa such as Dorea, Prevotella, Dialister invisus, Fusicatenibacter, and Coprococcus were associated with T2D, while beneficial taxa such as Eubacterium, Ligilactobacillus, and Acidaminococcus were more prevalent in healthy or prediabetic individuals.

Conclusions

This study reveals notable differences in the intestinal microbiota composition among patients with MS and T2D. Changes in microbial composition, particularly the Firmicutes/Bacteroidetes ratio, may serve as indicators of underlying pathology. At more specific taxonomic levels, several enriched taxa were identified in patients with MS, including Blautia, Tyzzerella, Dorea, Streptococcus, and Ruminococcus callidus. Additionally, species such as Dorea longicatena and Dialister invisus were enriched in prediabetic and diabetic patients, whereas beneficial genera (Eubacterium, Acidaminococcus, Bifidobacterium, and Ligilactobacillus) were more prevalent in healthy and prediabetic individuals than in those with T2D.

Introduction

Metabolic syndrome (MS) is a complex and prevalent clinical condition characterized by a cluster of metabolic abnormalities. These typically include central obesity, insulin resistance, dyslipidemia (elevated triglycerides and low high-density lipoprotein cholesterol), and elevated blood pressure (BP). Additionally, MS is associated with a prothrombotic state, proinflammatory conditions, non-alcoholic fatty liver disease, cholesterol gallstone disease, and reproductive abnormalities (Dommermuth & Ewing, 2018; Wang et al., 2020). The presence of MS significantly increases the risk of cardiovascular diseases and type 2 diabetes (T2D). Notably, several studies indicate that this condition is linked to an approximately twofold increase in cardiovascular disease risk and a fivefold increased risk of T2D (Grundy et al., 2005; Kathiresan et al., 2006; Després et al., 2008; Genser et al., 2016). Moreover, the prevalence of MS is rising globally—including in the United States, Europe, China, India, and South Korea—making it a major public health concern, with more than a quarter of the adult population estimated to be affected (Saklayen, 2018; Wang et al., 2020).

T2D is a chronic metabolic disease characterized by insufficient insulin levels and/or insulin resistance, leading to impaired glucose metabolism (Gomaa, 2020). It is marked by high blood glucose levels (hyperglycemia) and insulin resistance, and it accounts for approximately 90% of all diabetes cases worldwide (Sun et al., 2022). The hyperglycemic state, combined with environmental, genetic, dietary, and lifestyle factors, can lead to dysbiosis—an imbalance in the gut microbiota. In individuals with T2D, dysbiosis occurs when there is a mismatch between the Bacteroidetes and Firmicutes phyla, resulting in increased intestinal permeability. This allows bacterial by-products to pass through the intestinal membrane, triggering the inflammatory response characteristic of diabetes (Iatcu, Steen & Covasa, 2021). The rising prevalence of obesity—defined as a body mass index (BMI) of ≥30 kg/m2—further contributes to the development of several life-threatening non-communicable diseases, including MS, cardiovascular diseases, and T2D (Leitner et al., 2017).

Several studies have explored the association between the gut microbiota composition and MS/T2D. These investigations have highlighted specific microbial genera and species that may be either enriched or depleted in patients with these conditions (Qin et al., 2012; Karlsson et al., 2013; Wu et al., 2017).

MS and T2D are closely intertwined health challenges, with significant implications for global health. Understanding the role of the gut microbiome in these conditions has emerged as a promising avenue for future research, particularly in exploring its potential as a diagnostic marker and therapeutic target. With this in mind, we have conducted the following study, wherein we delve deeper into the specific genera and species within the gut microbiota associated with MS and T2D. We then compare and discuss our findings in relation to the current state of the art and present our conclusions.

Methods

Recruitment and volunteers

The volunteers for this study were part of the IBEROBDIA project (“Obesity and Diabetes in Iberoamerica: Risk Factors and New Pathogenic and Predictive Biomarkers”), funded by the Iberoamerican Program of Science and Technology (CyTED) (918PTE0540) and the Spanish State Research Agency (PCI2018-093284). This project was approved by the ethics committee of the Galician Health System (SERGAS, Xunta de Galicia), under code 2018/270. All participants were fully informed about the study and provided written informed consent. The volunteers’ data were processed in accordance with Organic Law 3/2018 of December 5 on the protection of personal data and the guarantee of digital rights. Participants were recruited through various media, including press, radio, and posters, in different locations across the autonomous community of Galicia, Spain.

The inclusion criteria for the study were Spanish adults aged 40 to 70 years, BMI of 18.5–24.9 kg/m2 (normal weight) or ≥27 kg/m2, and Finnish Diabetes Risk Score (FINDRISC) of ≥12, indicating a moderate risk of developing T2D. The exclusion criteria were consumption of prebiotic or probiotic supplements in the last 2 months, a prior diagnosis of T2D, the presence of other chronic diseases, pregnancy, antibiotic treatment within 2 months prior to the study, chronic medication use (including treatments for hypertension, high cholesterol, contraceptives, and proton pump inhibitors), drug consumption, and a low risk of developing diabetes according to the FINDRISC. Because chronic medication intake is known to influence the gut microbiota composition, it was a key exclusion criterion. Likewise, volunteers could not have a prior diagnosis of either T2D or MS.

Study design

This was a cross-sectional observational study conducted as part of the IBEROBDIA project, as described in the previous section. The study design is illustrated in the flowchart shown in Fig. 1. Following the provision of written informed consent, the volunteers were categorized based on BMI, sex, age, and their risk of developing T2D. Those who met the inclusion criteria were then invited to have their glucose levels checked and undergo an oral glucose tolerance test. On the day of the test, the volunteers were required to bring a fecal sample that had been collected at home for analysis.

Figure 1 Study design classification.

Based on the collected data, the participants were classified into three groups: individuals with normal weight and normal glucose values, individuals with a BMI of ≥27 kg/m2 and normal glucose values, and individuals with a BMI of ≥27 kg/m2 and altered glucose values. A second classification was then introduced based on BMI and the presence of MS, following the criteria established by the International Diabetes Federation in 2016 (Alberti, Zimmet & Shaw, 2016). Using these criteria, the groups were reorganized as follows: normal weight with MS, BMI of ≥27 kg/m2 with MS, and BMI of ≥27 kg/m2 without MS. Although individuals with a BMI of ≥27 kg/m2 are technically classified as overweight, they were included in the obesity group because their other characteristics—such as BP, biochemical markers, and glucose levels—closely resembled those of individuals in the obesity category with a BMI of ≥30 kg/m2.

Anthropometry

Height was measured using a portable stadiometer (Leicester Height Measure, Manchester, UK), and body composition was assessed using the In Body 127 scale (Microcaya S.L., Bilbao, Spain). Waist circumference was measured with a flexible measuring tape while volunteers wore light clothing without overgarments. Participants were then classified according to their BMI based on the criteria established by the Spanish Obesity Society (Salas-Salvadó et al., 2007).

Clinical parameters

The study volunteers were recruited early in the morning and instructed to fast for at least 8 h. Upon arrival at the laboratory, BP was measured following the procedure recommended by the American College of Cardiology/American Heart Association (Whelton et al., 2017). This protocol includes a 15-min rest before measurement, ensuring the patient has emptied their bladder, and avoiding conversation during both the rest period and the measurement, among other recommendations (Whelton et al., 2018).

For the measurements the Minimus® II manual sphygmomanometer (Riester, Jungingen, Germany) was used alongside with the Duplex® 2.0 phonendoscope (Riester, Jungingen, Germany). BP measurements were taken in duplicate, and BP was categorized into four levels based on the average values recorded, following the guidelines of the American College of Cardiology/American Heart Association: normal, elevated, and stage 1 or 2 hypertension (see Table 1).

Table 1 Blood pressure classification in adults.

BP category	Systolic blood presure		Diastolic blood preasure	
Normal	<120 mm Hg	and	<80 mm Hg	
Elevated	120–129 mm Hg	and	<80 mm Hg	
Hypertension				
Stage 1	130–139 mm Hg	or	80–89 mm Hg	
Stage 2	≥140 mm Hg	or	≥90 mm Hg	

After the BP measurements, a blood sample was taken from the volunteers to determine their plasma levels of glucose, high-density lipoprotein cholesterol, low-density lipoprotein cholesterol, very-low-density lipoprotein cholesterol, total cholesterol, triglycerides, insulin, and glycated hemoglobin. A capillary whole blood glucose measurement was then performed using a portable reflectance meter (OneTouch Select® Plus; Johnson & Johnson, S.A., Madrid, Spain). Volunteers with capillary glucose values of <126 mg/dL underwent an oral glucose tolerance test, which involved consuming a 75-g oral glucose load.

Criteria for volunteers’ classification: glucose levels and presence of MS

The classification of volunteers based on glucose levels was conducted using the standard diagnostic criteria proposed by the International Diabetes Federation and the World Health Organization (Magliano, Boyko & 10th Edition Scientific Committee, IDA, 2021). These criteria also incorporate recommendations from the American Diabetes Association, which defines “prediabetes” as HbA1c values between 5.7% and 6.4% and impaired fasting glucose as fasting plasma glucose levels between 100 and 125 mg/dL. These parameters were included in the classification criteria for the volunteers.

For the classification of volunteers with or without MS, we followed the International Diabetes Federation worldwide definition of MS (Alberti, Zimmet & Shaw, 2016), which includes: • Central obesity: Waist circumference—ethnicity specific (for Europeans: ≥94 cm for men and ≥80 cm for women). If BMI > 30 kg/m2, central obesity can be assumed.

• Plus, any of the following: – Raised triglycerides: ≥150 mg/dL

– Reduced high-density lipoprotein cholesterol: <40 mg/dL in men, <50 mg/dL in women

– Raised BP: systolic BP of ≥130 mmHg or diastolic BP of ≥85 mmHg

– Raised plasma glucose: fasting plasma glucose of ≥100 mg/dL or a diagnosis of T2D

Fecal DNA extraction and quantification

The commercial DNeasy PowerSoil Kit (Qiagen, Hilden, Germany) was used to extract bacterial DNA from the samples following the manufacturer’s instructions. The QubitTM 4 fluorometer (Invitrogen, Thermo Fisher Scientific, Waltham, MA, USA) and the DNA HS Assay Kit (Invitrogen, Thermo Fisher Scientific, Waltham, MA, USA) were used to quantify the extracted DNA from the samples. DNA samples were stored at −20 °C until further analysis.

16S rRNA amplicon sequencing

For 16S rRNA amplicon sequencing, 2 μL of DNA extracted from each sample was used to construct the libraries, and sequencing was performed using the Ion GeneStudioTM S5 system (Life Technologies, Carlsbad, CA, USA). The 16S hypervariable regions were amplified with two primer sets, V2-4-8 and V3-6,7-9, and libraries were prepared using the Ion 16STM Metagenomics Kit (Life Technologies, Carlsbad, CA, USA) and the Ion XpressTM Plus Fragment Library Kit (Life Technologies, Carlsbad, CA, USA). Libraries containing equal amounts of polymerase chain reaction products combined with a barcode were prepared using the Ion XpressTM Kit barcode adapters (Life Technologies, Carlsbad, CA, USA) and quantified using the Ion Universal Library Quantification Kit (Life Technologies, Carlsbad, CA, USA). Next, 10 pM of each library was pooled and loaded onto an Ion OneTouchTM 2 system (Life Technologies, Carlsbad, CA, USA), which automatically performed template preparation and enrichment. Positive ion sphere particles for the template were enriched using DynabeadsTM MyOneTM Streptavidin C1 magnetic beads (Invitrogen, Carlsbad, CA, USA) with an Ion OneTouchTM ES instrument. Finally, an Ion 520TM chip (Life Technologies, Carlsbad, CA, USA) was loaded with the samples on an Ion GeneStudioTM S5 System sequencer, using the Ion 520TM and Ion 530TM loading reagents supplied in the OT2 Kit (Life Technologies, Carlsbad, CA, USA). Portions of this text were previously published as part of our prior work (Sinisterra-Loaiza et al., 2023).

Bioinformatics analyses

For all 16S rRNA sequence data, the quality of the raw reads was visualized using FastQC v0.11.7 (Andrews, 2010) to ensure an average quality score of at least 25. The reads were then imported into R (v4.2.0) (R Core Team, 2013) and assembled into amplicon sequence variants (ASVs) using the DADA2 package (v1.26) (Callahan et al., 2016). To asses the length distribution of all sequences in aggregate terms a second quality check of the samples was performed using the quality plots provided by DADA2.

For sequence processing, the default parameters provided by the author of DADA2 were left unchanged. However, following the author’s recommendations for sequences from Ion Technologies, several parameters were modified. In the initial filtering phase, a trimLeft value of 15 was applied. During the inference of real biological sequences phase, a homopolymer gap penalty of −1 and a band size of 32 were set. A total of 20,892 unique variants reached the chimera elimination phase. Of these, 9.9% were removed after being identified as chimeras by the algorithm, leaving 18,831 unique sequences. However, these chimeras accounted for only 1.5% of the total abundance of variants. The non-chimeric assembled ASVs were then taxonomically assigned (from phylum to species) using the Silva reference database (v138.1) (McLaren & Callahan, 2021).

The ASVs table, taxonomic table, and clinical data of the patients were combined into a phyloseq object (v1.42.0) (McMurdie & Holmes, 2013) for downstream bioinformatics analysis. Once the phyloseq object was constructed, agglomeration was performed at different taxonomic levels, followed by an initial filtering phase. During this phase, taxa that were not present more than three times in at least 20% of the samples were eliminated. Additionally, taxa with fewer than 20 counts were removed to ensure data reliability.

The phyloseq package was used to calculate bacterial diversity, including both alpha-diversity and beta-diversity. Alpha-diversity was evaluated using the Shannon index and the Simpson index, while beta-diversity was assessed using the Bray–Curtis index and the Chao index. To visually examine group dissimilarities, principal coordinate analysis (PCoA) was applied. Alpha-diversity was analyzed using non-parametric statistical tests, specifically the Wilcoxon test for comparisons between two groups and the Kruskal–Wallis test for comparisons among three or more groups, implemented through the ggpubr package (v0.5.0) (Kassambara & Kassambara, 2020). Beta-diversity was assessed using permutational analysis of variance with 1,000 permutations, performed with the adonis2 function from the R vegan package (Oksanen et al., 2022).

The LinDA method (Zhou et al., 2022) was used to measure the differences in abundance between the various groups. This method was specifically designed for differential abundance analysis of microbiome compositional data. When calculating differences in the abundance of genera and species, BMI was used as a covariate to ensure that the model accounted for data from both normal-weight and obese patients, thereby minimizing potential bias. The MicrobiomeStat package (Zhang & Chen, 2022) was used to perform this analysis.

Results

In total, 79 volunteers met all recruitment criteria and provided a fecal sample. The classification based on glucose levels was as follows: individuals with normal weight and normal glucose values (n = 23), individuals with obesity and normal glucose values (n = 35), and individuals with obesity and altered glucose values (n = 21). The categorization according to BMI and the presence of MS was as follows: individuals with normal weight and MS (n = 2), individuals with obesity and MS (n = 22), and individuals without MS (n = 55).

At the conclusion of the study, for downstream analysis, 55 individuals were identified as having MS while 22 did not exhibit this condition. Additionally, 59 individuals were classified as non-diabetic, 15 as pre-diabetic, and five as diabetic. As previously discussed, these analyses incorporated prior classifications (such as BMI and glucose status) as covariates to minimize potential bias. Table 2 summarizes the data obtained for volunteers classified by BMI.

Table 2 Summary of study patient characteristics.

Variable	Individual with normal weight (NW)	Individual with obesity (OB)	
Recruited volunteers	n = 22	n = 57	
Men	n = 11	n = 22	
Women	n = 11	n = 35	
Age (years)	52.00 ± 6.53	52.47 ± 7.49	
Height (cm)	164.00 (1.60–180.00)	165.43 ± 9.42	
Weight (kg)	64.20 ± 7.49	86.57 ± 13.81	
BMI (kg/m2)	23.12 ± 1.40	31.10 (27.20–41.90)	
Waist circumference (cm)	86.68 ± 8.38	103.47 ± 9.2	
Fasting glucose (mg/dL)	86.32 ± 6.75	93.00 (65.00–298.00)	
Glucose at 2 h (PTOG) (mg/dL)	78.88 ± 17.80	104.76 ± 26.01	
Systolic Blood Pressure (SBP; mmHg)	113.41 ± 15.69	125.37 ± 15.41	
Diastolic Blood Pressure (DBP; mmHg)	71.41 ± 9.90	75.00 (60.00–100.00)	
Hb1Ac	4.92 ± 0.24	5.00 (4.40–11.00)	
Insulin ( μUl/mL)	6.60 ± 2.98	8.25 (4.30–77.10)	
HOMA-IR	1.44 (0.68–3.88)	2.04 (0.96–14.07)	
Triglycerides (mg/dL)	76.95 ± 29.66	101.00 (38.00–319.00)	
HDL-c (mg/dL)	3.34 ± 0.96	3.99 ± 0.92	

The normality of numerical variables was assessed using the Anderson–Darling test via the nortest package Gross & Ligges (2015) in R. For variables that did not follow a normal distribution, data are reported as median (min–max), while normally distributed variables are presented as mean ± standard deviation.

Microbial composition and diversity in MS and T2D

In the initial phase, a series of exploratory analyses were conducted to examine the overall microbial composition and diversity in relation to MS and T2D. These analyses considered both the entire population and different pathological conditions. Figure 2A illustrates the abundance of various taxa across different taxonomic levels. The relative abundance of up to 10 taxa is represented individually, with the remaining taxa aggregated. At the phylum level, Firmicutes and Bacteroidota were the most abundant, followed by Actinobacteriota and Proteobacteria. At the family level, Lachnospiraceae and Bacteroidaceae dominated the microbiota of this cohort, followed by Ruminococcaceae and Prevotellaceae, with abundance gradually decreasing across the remaining families. Finally, at the genus level, the microbiota was primarily characterized by the abundance of the Bacteroides genus, followed by a substantial number of genera with more homogeneous abundances.

Figure 2 Exploratory analysis of the cohort.

(A) Abundance across different taxonomic levels, including Phylum, Family, and Genus. (B) Comparison of Phylum ratios between healthy individuals and patients with metabolic syndrome. (C) Comparison of Phylum ratios among healthy individuals, patients with prediabetes, and patients with diabetes. (D) Alpha diversity (Shannon and Simpson indices) and beta diversity (Bray and Chao distances) at the genus level among healthy individuals and patients with metabolic syndrome. (E) Alpha diversity (Shannon and Simpson indices) and beta diversity (Bray and Chao distances) at the genus level among healthy individuals, prediabetic patients, and diabetic patients.

Subsequently, the ratios of different phyla were analyzed in relation to MS and T2D (all data available in File S1). In patients with MS (Fig. 2B), the relative abundance of the Firmicutes phylum was higher than in healthy individuals, along with an increase in Bacteroidetes. The proportions of the remaining phyla remained similar between both groups. For T2D, the relative abundance of Firmicutes and Actinobacteriota decreased as the disease progressed, while an inverse trend was observed for Bacteroidetes. Additionally, the relative abundance of Proteobacteria was slightly higher in diabetic individuals than in healthy controls. The proportions of the other phyla remained relatively stable across groups, as shown in Fig. 2C.

Alpha diversity, assessed using the Shannon and Simpson indices (Fig. 2D), was first evaluated to detect potential differences in microbial diversity between patients with MS and healthy controls. The results showed no statistically significant differences, suggesting that the overall microbiota diversity in patients with MS is similar to that of healthy individuals. For beta diversity, the Bray–Curtis and Chao distance metrics were used to assess differences in microbial community composition between the two groups. The PCoA plot in Fig. 2D reveals a significant separation between patients with MS and healthy controls, with associated p-values of 0.002 and 0.01. This indicates a marked compositional shift in the microbiota between these groups despite the similarity in alpha diversity.

The second experiment aimed to compare microbiota diversity among healthy individuals, prediabetic patients, and diabetic patients. Alpha diversity, evaluated again using the Shannon and Simpson indices (Fig. 2E), did not reveal significant differences between these groups, indicating comparable microbial diversity across the spectrum. The PCoA plots in Fig. 2E illustrate no discernible differences in microbiota composition among healthy individuals, prediabetic patients, and diabetic patients. Additionally, statistical analysis did not yield significant p-values, suggesting that microbiota compositional changes between these groups are not statistically robust.

Analysis of taxon abundances in MS

Subsequently, we aimed to examine variations among taxa across patient groups and identify specific patterns within these taxa. To achieve this, an abundance difference analysis was conducted using the LinDA method. Additionally, the analysis was performed at both the genus and species levels to assess differences based on taxonomic depth. Detailed results of these analyses are available in File S1.

Figure 3A presents the results of the comparison between healthy individuals and patients with MS. Upon careful examination of the figure, it became evident that genera such as Tyzerella, Streptococcus, Blautia, Family XIIIAD3011 group, Dorea, and Paludicola, among others, were enriched in patients with MS, with log2 fold changes (log2FCs) ranging from approximately 1.5 to 3.0. By contrast, healthy individuals exhibited a higher relative abundance of genera such as [Eubacterium] eligens group and Prevotellaceae NK3B31 group, with log2FCs lower than −2.

Figure 3 Volcano plots comparing microbiome abundances between healthy individuals and patients with metabolic syndrome.

Red dots represent taxa with p-values less than 0.05 and |log2FCs| greater than 1. (A) Comparison of abundances between healthy individuals and metabolic syndrome patients at genus level. (B) Comparison of abundances between healthy individuals and metabolic syndrome patients at species level.

At the species level (Fig. 3B), the representation of the genus Blautia in patients with MS remained and became more pronounced, with four species—Blautia faecis, Blautia massiliensis, Blautia obeum, and Blautia wexlerae—showing higher relative abundance. The genus Dorea also maintained significance, represented by Dorea formicigenerans. These species were accompanied by others such as Bifidobacterium bifidum, Intestinibacter bartlettii, Roseburia intestinalis, and Ruminococcus callidus, with log2FCs ranging from approximately 1.5 to 3.0. On the other hand, healthy individuals appeared to be enriched in the species Parabacteroides merdae, which exhibited a log2FC of approximately −2.

Analysis of taxon abundances in T2D

Next, to better understand the role of the microbiota in T2D, differences in taxon abundances were analyzed across three comparisons: healthy individuals vs. prediabetics, healthy individuals vs. diabetics, and prediabetics vs. diabetics. As in the previous case, detailed results of these analyses are available in File S1.

In the comparison between healthy individuals and those with prediabetes, Fig. 4A illustrates that the genus Dorea ( p = 7.80535×10−9, log2FC = 10) was highly enriched in prediabetic patients. Additionally, genera such as Anaerostipes, Coprococcus, Fusicatenibacter, Subdoligranulum, and [Eubacterium] hallii group, among others, showed increased abundance in prediabetic individuals, with log2FCs between 7.5 and 10. By contrast, genera such as Ligilactobacillus, Acidaminococcus, Bifidobacterium, Faecalibacterium, Collinsella, Defluviitaleaceae UCG-011, and GCA-900066575, among others, were enriched in healthy individuals, with log2FCs ranging from −5 to −15.

Figure 4 Volcano plots comparing the abundance of the microbiome between healthy individuals, pre-diabetic patients and diabetic patients.

Red dots represent taxa with p-values less than 0.05 and |log2FCs| greater than 1. The first column shows comparisons at the genus level, while the second column presents comparisons at the species level. (A) Comparison between healthy individuals and patients with pre-diabetes. (B) Comparison between healthy individuals and diabetic patients. (C) Comparison between pre-diabetic and diabetic patients.

At the species level, four genera enriched in prediabetic patients persisted as the only species showing higher relative abundance in this group. These species were Dorea longicatena, Anaerostipes hadrus, Fusicatenibacter saccharivorans, and Coprococcus comes, with log2FCs ranging from approximately 5 to 8. In healthy individuals, species from three genera were enriched, represented by Lachnoclostridium edouardi, Collinsella aerofaciens, and Bifidobacterium bifidum, with log2FCs ranging from approximately −5 to −8.

In the comparison between healthy individuals and diabetics, shown in Fig. 4B, the genus Dorea ( p = 2.16796×10−5, and a log2FC = 7.5) appeared to be more dominant in diabetic patients, followed by Anaerostipes and Subdoligranulum. Meanwhile, in healthy individuals, genera such as Ligilactobacillus, Acidaminococcus, Bifidobacterium, Faecalibacterium, GCA-900066575, Defluviitaleaceae UCG-011, [Eubacterium] ventriosum group, and Collinsella, among others, exhibited greater prevalence, with log2FCs ranging from approximately −5 to −12.5.

At the species level, a similar pattern to that observed at the genus level was evident. The only species enriched in diabetic patients was Dorea longicatena, with a log2FC of approximately 5. In contrast, in healthy individuals, species such as Alistipes putredinis, Bifidobacterium adolescentis, Bifidobacterium longum, Blautia faecis, Butyricicoccus faecihominis, and Collinsella aerofaciens were enriched, with log2FCs ranging from approximately −5 to −10.

Finally, Fig. 4C, illustrates the differences between prediabetic and diabetic patients. In this case, genera such as Prevotella, Dialister, and Sutterella were more dominant in diabetic patients, with log2FCs ranging from 5.0 to 7.5. On the other hand, genera such as CAG-56, Erysipelotrichaceae UCG-003, Fusicatenibacter, Lachnospiraceae FCS020 group, Roseburia, [Eubacterium] hallii group, [Eubacterium] ventriosum group, and Dorea, among others, were more prevalent in prediabetic individuals, with log2FCs ranging from −2.5 to −5.0.

At the species level, diabetic patients exhibited an enriched abundance of Butyricimonas faecihominis, Dialister invisus, Lachnoclostridium edouardi, and Odoribacter splanchnicus, with log2FCs ranging from approximately 2.5 to 5.0. By contrast, species such as Blautia faecis and Dorea longicatena were enriched in prediabetic patients, with log2FCs ranging from approximately −2 to −4.

Discussion

Following acquisition of the above results, an exhaustive review of the scientific literature was conducted to contextualize and compare the findings of this study with previously reported data in specialized research.

In patients diagnosed with MS, a higher Firmicutes/Bacteroidetes ratio was observed. Notably, several studies have supported this phenomenon, consistently reporting an elevated proportion of Firmicutes in individuals with MS (Santos-Marcos, Perez-Jimenez & Camargo, 2019; Baghi et al., 2022).

In the population of individuals with MS in our cohort, a notable observation is the consistently high prevalence of the genus Blautia. This genus plays a central role in the fermentation of carbohydrates and other substrates. It produces short-chain fatty acids (SCFAs) such as acetic acid, propionic acid, and butyric acid, which are crucial for the health of the intestinal epithelium (Liu et al., 2021b). While some studies suggest an increased abundance of Blautia in individuals with MS, the underlying mechanisms of this association remain unclear. Previous research has indicated that a greater presence of Blautia may contribute to metabolic dysfunction through pathways such as inflammation and intestinal barrier disruption (Liu et al., 2021b). Studies analyzing the gut microbiota of 85 individuals with MS and 58 healthy controls have confirmed a significant increase in Blautia abundance in patients with MS compared with healthy controls. Furthermore, a positive correlation has been identified between Blautia abundance and MS severity in patients with non-alcoholic fatty liver disease, a condition closely related to MS (Shen et al., 2017; Wang et al., 2016).

However, it is important to note that contradictory findings have been reported in the scientific literature. While some studies have linked an increase in Blautia to conditions such as type 1 diabetes in children, obesity, and chronic kidney disease (Kostic et al., 2015; Pataky et al., 2016; Barrios et al., 2015), other reports indicate a significant decrease in Blautia in patients with type 1 and T2D, showing an inverse correlation with glycated hemoglobin and plasma glucose levels (Murri et al., 2013; Inoue et al., 2017). A reduction in Blautia has also been observed in cases of systemic inflammation and insulin resistance (Benítez-Páez et al., 2020). Given this complex variability, it is concluded that while the abundance of Blautia is closely associated with these pathologies, the precise and direct relationship between Blautia, health, and disease has not yet been fully elucidated in the current scientific literature.

Regarding other enriched taxa in patients with MS, the genus Tyzerella has been observed to exhibit a notable predominance in gestational diabetes, with a positive correlation identified with fasting blood glucose levels (Ma et al., 2020). Additional studies support the association of Streptococcus with MS (Aran et al., 2011). On the other hand, the species Ruminococcus callidus has been linked to various conditions, including colorectal cancer, gestational diabetes, and irritable bowel syndrome (Osman et al., 2021; Ye et al., 2023; Cremon et al., 2018). The species Dorea formicigenerans has been associated with metabolic disorders and obesity (Brahe et al., 2015; Companys et al., 2021).

Finally, in our study, an increase in the species Parabacteroides merdae was observed in healthy individuals. While specific information about this species is limited in the literature, previous studies have reported a decrease in certain species within the genus Parabacteroides in individuals with MS (Cui et al., 2022).

However, an inconsistency has been identified in the case of Roseburia intestinalis. While the present study found a higher prevalence of Roseburia intestinalis in individuals with MS, previous research has linked this species to health benefits and the promotion of a healthier metabolic state, as supported by multiple bibliographic sources (Hur & Lee, 2015; Nie et al., 2021; Qin et al., 2012). Finally, no notable findings have been identified regarding the taxa Paludicola, Family XIIIAD3011 group, or Intestinibacter bartlettii.

In the context of diabetes, patients with diabetes have been found to exhibit a decrease in the relative abundance of phyla such as Actinobacteria and Firmicutes, along with a notable reduction in the Firmicutes/Bacteroidetes ratio compared with non-diabetic patients. Additionally, an increase in the relative abundance of bacteria from the phylum Bacteroidetes has been observed in diabetic patients. Notably, a significant rise in the presence of pathogenic taxa, such as Proteobacteria and Bacteroidetes, has been recorded in these patients. These patterns align with previous research, which has identified similar microbial shifts in both T2D and other forms of diabetes (Larsen et al., 2010; Murri et al., 2013; Fassatoui et al., 2019; Su et al., 2022).

Regarding enriched taxa in prediabetic and diabetic patients, little is known about the specific role of the genus Dorea in the intestinal microbiota. It has been demonstrated that Dorea thrives in the presence of a diet rich in fermentable carbohydrates, enabling it to produce lactic acid. Some studies have suggested that Dorea may be associated with obesity and altered glucose levels, as its higher abundance has been observed in individuals with T2D compared to those without (Zhang et al., 2021). At the genus level, Dorea has also been linked to more complex conditions, such as colorectal cancer (Yang et al., 2019).

In the specific case of Dorea longicatena, this species has been found to be enriched in patients with T2D. A significant clinical correlation has been observed between its presence and the development of the disease (Qin et al., 2014). It has also been enriched in patients with obesity and metabolic disorders (Liu et al., 2017). One possible explanation for the increase in Dorea in individuals with prediabetes and T2D is its role in the degradation of complex carbohydrates. This process leads to a higher production of SCFAs, which are bacterial metabolic byproducts that can influence glucose metabolism and host insulin sensitivity. For example, propionic acid, upon reaching the liver, is utilized for glucose production, and an increase in this byproduct may contribute to a slight elevation in blood glucose levels. Additionally, the low-grade systemic inflammation present in these individuals due to obesity may alter intestinal microbiota composition, promoting the increase of certain bacterial species, such as Dorea. These findings suggest that the presence or abundance of Dorea longicatena in the intestinal microbiota may impact glucose metabolism and insulin regulation, potentially influencing the development and progression of T2D.

With respect to Prevotella, several studies have established an association between an increase in this bacterium and impaired glucose tolerance, as well as gestational diabetes (Zhang et al., 2013; Egshatyan et al., 2016; Hasain et al., 2020). On the other hand, the species Dialister invisus has been linked to increased intestinal permeability and diabetes (Maffeis et al., 2016; Zheng, Li & Zhou, 2018). Similarly, the genus Sutterella has been associated with obesity and has shown a positive correlation with T2D and potential complications in these patients (Squillario et al., 2023; Gradisteanu Pircalabioru et al., 2022). These taxa exhibit higher representation in individuals with diabetes compared to those in a prediabetic state, suggesting that both taxa could serve as indicators of disease progression.

Regarding Fusicatenibacter, the available data are limited and contradictory. Some studies suggest its association with inflammatory bowel diseases, colorectal cancer, and other metabolic disorders, while others have reported a significant decrease in Fusicatenibacter in individuals with MS compared to healthy individuals. In our study, we observed a higher abundance of Fusicatenibacter in prediabetic volunteers compared with the other two groups, suggesting a potential relationship with the pathogenesis or early stages of diabetes.

As for Anaerostipes, similar to Fusicatenibacter, contradictory findings are reported in the literature. Some studies at the genus level suggest a positive correlation between the presence of Anaerostipes and metabolic health in diabetic patients (Doumatey et al., 2020). However, research at the species level indicates that Anaerostipes hadrus is positively correlated with the presence of diabetes (Liu et al., 2021a). In our study, individuals with prediabetes and diabetes exhibited a higher abundance of the genus Anaerostipes than did healthy individuals.

In multiple studies, an increase in the species Coprococcus comes has been associated with metabolic disorders and gestational diabetes mellitus (Guo et al., 2018; Dualib et al., 2021). However, the higher representation of Coprococcus in these individuals could be attributed to its ability to produce SCFAs, which are beneficial in regulating blood glucose levels and glucose uptake. Moreover, Coprococcus spp. may play a role in the metabolism of folate and biotin, vitamins that have been linked to lower plasma glucose levels. Therefore, the abundance of this taxon in patients with altered glucose levels, such as those with T2D, could be related to its function in modulating elevated glucose levels (Den Besten et al., 2013; Tettamanzi et al., 2021).

Regarding Subdoligranulum, studies in humans with T2D have reported a lower abundance of this bacterium than in individuals without the disease (Liu et al., 2018). However, in our study, this taxon exhibited a higher abundance in volunteers in the prediabetes stage and those with T2D. This finding aligns with previous research linking Subdoligranulum to insulin resistance, as indicated by the HOMA-IR model. It is known that this bacterium produces acetic and butyric acids, with butyrate playing a key role in reducing intestinal pH and enhancing intestinal calcium absorption, which in turn may decrease insulin resistance (Shumar, 2019).

Upon detailed analysis of enriched taxa in non-diabetic individuals, the genus Eubacterium exhibits higher abundance in healthy or prediabetic patients compared to those with T2D. Numerous previous studies have highlighted the beneficial role of this genus in metabolic health, supporting our observation, and have noted its higher prevalence in individuals without diabetes (Pushpanathan et al., 2016; De Moraes et al., 2017; James et al., 2022; Wu et al., 2021). Additionally, in our study, two species of Eubacterium have been identified that follow this trend, further strengthening the association with metabolic health.

On the other hand, Roseburia appears to play a positive role in the health of patients with T2D, as suggested by previous research (Murphy et al., 2017; Ejtahed et al., 2020). Although our study did not find significant differences in Roseburia abundance between healthy individuals and prediabetics, we observed a notable difference between prediabetics and diabetics, with a higher abundance in the prediabetic group.

Our study has provided insights into the composition of the intestinal microbiota in relation to metabolic health, identifying several enriched taxa in healthy individuals, suggesting a potential role in preventing metabolic diseases. Among these findings, the genus Ligilactobacillus stands out. Its predominant presence in healthy subjects aligns with previous research in animal models, where its beneficial role in glucose regulation and metabolic homeostasis has been demonstrated (Peng & Zheng, 2023; Liang et al., 2021).

On the other hand, the genus Acidaminococcus has been associated, in a singular study, with a slightly lower risk of developing T2D (Yang et al., 2018). Furthermore, in line with previous research, we have confirmed a significant decrease in the abundance of Bifidobacterium in patients with T2D compared with individuals without the disease (Larsen et al., 2010; Wu et al., 2010; Lê et al., 2012). Notably, the therapeutic potential of certain Bifidobacterium species, such as Bifidobacterium adolescentis, Bifidobacterium longum, and Bifidobacterium bifidum, is actively being investigated for their role in improving insulin resistance and glucose tolerance (Zhao et al., 2020; Qian et al., 2022; Hao et al., 2022).

Additionally, in line with previous research linking the genus Blautia to diabetes pathology (Murri et al., 2013; Inoue et al., 2017), we observed a decrease in the abundance of Blautia faecis with T2D compared with both healthy individuals and prediabetics, further supporting these previously reported associations. Finally, the species Alistipes putredinis appears to be positively correlated with better glucose tolerance (Davies et al., 2020; Wu et al., 2020; Sun et al., 2023; Ding et al., 2023).

The species Collinsella aerofaciens has shown a positive association with healthy individuals in our study, despite previous research linking it to higher abundances in individuals with T2D (Kulkarni, Devkumar & Chattopadhyay, 2021; Wu & Park, 2022; Demirci et al., 2022). These contrasting findings suggest that the role of Collinsella aerofaciens in metabolic health may be more context-specific than previously understood, potentially influenced by environmental or host factors. Such discrepancies highlight the need for further investigation into the complex, bidirectional relationships between the gut microbiome and metabolic disease.

These findings underscore the complexity of the intestinal microbiota and its potential role in metabolic health, emphasizing the need for further research to deepen our understanding of these interactions and explore potential clinical applications.

Importantly, this study has certain limitations that should be considered when interpreting the results. Firstly, the cohort size of 79 individuals may be relatively small in the context of clinical research. Additionally, we observed a significant imbalance between the prediabetic and diabetic groups compared with healthy subjects, which could impact the robustness of the obtained values. This disparity in class sizes might influence the ability of our methodology to detect significant differences, potentially affecting the resolving power of our conclusions. These limitations emphasize the need for future studies to replicate our findings in larger and more balanced cohorts to achieve a more comprehensive understanding of the microbial relationships with the studied pathologies.

Conclusions

This study contributes to existing knowledge by describing associations between microbial compositions and complex health conditions such as MS and T2D. The results obtained from this cohort of 79 patients revealed that even at a high taxonomic level, such as the phylum level, changes in microbial composition (particularly in the Bacteroidetes/Firmicutes ratio) may serve as indicators of pathology. These findings became more pronounced at the genus and species levels, where differences in various taxa were identified, further supported by the current state of the art.

Taxa such as the genus Blautia, followed by others like Tyzzerella, Dorea, Streptococcus, Ruminococcus callidus, and Dorea formicigenerans, were found to be enriched in patients with MS. Additionally, Parabacteroides merdae was identified as enriched in healthy individuals, suggesting a potential protective role, as species within this genus appear to exhibit such capabilities.

Furthermore, taxa from the genera Dorea, Prevotella, and Dialister were positively correlated with T2D. Species such as Dorea longicatena, Anaerostipes hadrus, Fusicatenibacter saccharivorans, and Dialister invisus, among others, were enriched in prediabetic or diabetic patients. By contrast, genera such as Eubacterium, Acidaminococcus, Bifidobacterium, and Ligilactobacillus, which appear to play a beneficial role in metabolic health, were enriched in healthy individuals and in prediabetic patients compared to those with T2D.

Supplemental Information

Supplemental Information 1 Experimental data (ratios, p-values, adjusted p-values, etc).

Additional Information and Declarations

Competing Interests

Carlos Fernandez-Lozano is an Academic Editor for PeerJ.

Author Contributions

Laura Isabel Sinisterra Loaiza conceived and designed the experiments, performed the experiments, authored or reviewed drafts of the article, and approved the final draft.

Diego Fernández-Edreira analyzed the data, prepared figures and/or tables, authored or reviewed drafts of the article, and approved the final draft.

Jose Liñares-Blanco analyzed the data, prepared figures and/or tables, authored or reviewed drafts of the article, and approved the final draft.

Alberto Cepeda conceived and designed the experiments, performed the experiments, authored or reviewed drafts of the article, and approved the final draft.

Alejandra Cardelle-Cobas conceived and designed the experiments, performed the experiments, authored or reviewed drafts of the article, and approved the final draft.

Carlos Fernandez-Lozano analyzed the data, authored or reviewed drafts of the article, and approved the final draft.

Human Ethics

The following information was supplied relating to ethical approvals (i.e., approving body and any reference numbers):

This project has the approval of the ethics committee of the Galician Health System (SERGAS, Xunta de Galicia), code 2018/270. All the participants included in the study were adequately informed about the process and signed an informed consent. Data from volunteers were processed in accordance with the organic law 3/2018, of 5 December, on the protection of personal data and the guarantee of digital rights.

The participants in this study were recruited through the dissemination of the project in different media, press, radio, posters, etc. in different locations of from the autonomous community of Galicia, Spain.

Data Availability

The following information was supplied regarding data availability:

The source code to reproduce all the analysis, along with documentation, is available on GitHub and Zenodo:

- https://github.com/MALL-Machine-Learning-in-Live-Sciences/IBEROBDIA.

- Diego FE, & Carlos Fernandez-Lozano. (2024). MALL-Machine-Learning-in-Live-Sciences/IBEROBDIA: 0.0.1 (0.0.1). Zenodo. https://doi.org/10.5281/zenodo.14051704.

The raw data in phyloseq format is available at Figshare: Fernandez Edreira, Diego; Fernandez-Lozano, Carlos; Liñares, Jose (2025). Data. figshare. Dataset. https://doi.org/10.6084/m9.figshare.26063020.v1.

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
