# Peer review of "Fecal microbiome analysis in patients with metabolic syndrome and type 2 diabetes"

_PeerJ, doi:10.7717/peerj.19108_

## Round 0.1 · original submission · Major Revisions

· Academic Editor

Major Revisions

The manuscript title “Study of the faecal microbiome of patients with metabolic syndrome and type 2 diabetes” is not perfectly aligned with the study subjects’ selection criteria as “diagnosis of T2D” is one of the exclusion criteria. The authors may highlight the reasons behind the set inclusion/exclusion criteria to make the readers understand the logic for using the specified selection criteria.
The manuscript requires major revision according to Reviewers’ comments, particularly in enhancing the clarity of text; re-writing of results in more effective and technical way; detailed explanation of figure legends; and evidence-based concluding remarks. Of course, correction of grammar and punctuation errors is crucially important.

Reviewer 1 ·

Basic reporting

Abstract:
Your abstract needs more detail. I suggest explaining more about the methods, such as the cohort period, group classification, and subject criteria.

Language Improvement:
The English language should be improved. In the results section, "higher" in line 29 should be followed by a comparison term. I also noticed inconsistency in the abbreviation for type 2 diabetes mellitus. Previously, you used "T2D," but in the results section, you used "DT2."

Introduction:
In line 56, I suggest using a more appropriate sentence. For example: "T2D is a chronic metabolic disease characterized by insufficient insulin levels and/or insulin resistance, leading to impaired glucose metabolism."

Methods:
Please check your numbering. There are several duplicate numbers in the methods section. For example, lines 90 and 91 have the same number.

Study Design:

I suggest providing more details about the study design, such as the type of cohort, the number of participants included in this research, and the subject inclusion criteria.
I suggest adding a flowchart to make it easier to understand.
Please add the statistical analysis methods you chose.
Results:

Table 2 only explains the characteristics of individuals with normal weight versus individuals with obesity. However, in the methods and results sections, it previously explains that the group is divided into three parts (normal weight with normal glucose vs. obesity with normal glucose vs. obesity with altered glucose values).
There is some inconsistency in the results. In Figure B, the subjects are divided into two groups (no MetS vs. MetS), but in the same figure, the subjects are divided into three groups (healthy vs. pre-DT2 vs. DT2).
Please check for any misstyping.

Experimental design

Study design is not well explained. It makes readers confuse regarding the design. It is a categorical observational research or comparisson categorical research or ? is it cohort prospective ? please explain period of time.

Subjects exclusion:
1. line 91: Chronic medication (hypertension, cholesterol, contraceptives, PPI and etc...). Based on this statement I suggest that you use new onset metabolic syndrome patients as your research subject. Because mostly this drugs is taken by the MetS patients. Please elaborate more.
2. Line 91: Please elaborate on the difference between antibiotic consumption in the last two months versus antibiotic and probiotic treatment in the past two months. Are they the same? If not, please clarify.
3. Please explain the criteria for volunteers versus participants in your paper. In participants, there is no MetS criteria, but in volunteers, there is a MetS criteria.

Validity of the findings

Data from table with abnormal distribution should report as: if the data is numerical, please use median (min-max). I found there is several abnormal data distribution in your table data. Please check again.

Reviewer 2 ·

Basic reporting

The abstract is too long with many vague and meaningless sentences. For example,

“Our study provides a detailed insight into the composition of the intestinal microbiota in relation to MS and T2D. Despite its smaller size, this non-homogeneous Galician cohort has allowed us to validate a set of microbial genera and species that, a priori, were associated with these metabolic diseases. These findings not only contribute to our understanding of the microbiota in the context of metabolic health but also can serve as a foundation for future research and the development of potential biomarkers, with promising applications in the field of metabolic health.”

Above is the “conclusion” part of the abstract, despite of its length, it contains no findings or any meaningful messages.

The manuscript, especially the METHODS part, is filled with numerous grammar errors. For examples,

Line 95, “After signed the informed consent”

Line 96, “After, those who met the inclusion criteria were invited to check your glucose levels”

Line 97, “The same day of the test they should be delivered a fecal sample collected at home”. (why the volunteers should be delivered a fecal sample?)

Line 116, “by the by the Amcerican College of Cardiology...”

Line 103, “Considering these criteria the groups were reordered into was...”

There are many places where commas or spaces are missing or misuse of symbols.

There are many misuses of the words “according to”.

Experimental design

Line 207-208, “i) individuals with normal weight and MS (n=2), ii) individuals with obesity and MS (n=22), and iii) individuals without MS (n=55)”. The sample size of different groups are extremely imbalanced, especially that there are only two samples for the obesity & MS group. Although non-parametric statistical tests can still be performed, this experimental design is by no means reasonable (as noted by the authors, line 453-455).

Line 105, individuals with “BMI=27” is classified as “with obesity”. I do not think that is reasonable.

Validity of the findings

Line 223-224, “Shifting attention to T2D, the proportion of Firmicutes and Actinobacteriota decreases as the diabetes stage progresses”. First, the diabetes stages cannot be simplified by a health-prediabetic-diabetic classification. Second, when you say “increase” or “decrease”, it needs to be statistically significant, which is missing in this context.

In the legend of Figure 2, “A) Difference in abundances between both groups at the genus level. B) Difference in abundances between both groups at the species level”—it does not indicate what are the “both groups”. Besides, the p-values need to be adjusted for multiple comparison correction.

The authors misuse the words “increase” or “decrease” very often. For example, line 339-340, “an increase in the species Parabacteroides merdae has been observed in healthy individuals.”—I think what the authors want to say is that the relative ratio of the species Parabacteroides merdae in healthy individuals’ gut microbiome is higher than MS patients, i.e., it does not make sense to simply say something increases.

There are many vague statements that do not seem very scientific. For examples, line 441-444,

“the species Colinsella aerofaciens has shown a positive correlation with healthy individuals, despite previous studies associating it with higher abundance in
individuals with T2D (Kulkarni et al., 2021; Wu and Park, 2022; Demirci et al., 2022). This finding raises significant questions and highlights the complexity of microbial relationships in the context of T2D.”

I do not see how that correlation “raises significant questions and highlights the complexity of microbial relationships in the context of T2D.”, i.e., what are the significant questions and how does it highlight the complexity of microbiome?

I do not understand why the authors start with “To conclude” (line 450) when mentioning this work’s limitations.

As the conclusion, line 461-462, “This study contributes to the existing knowledge on microbial composition and its effects on current and complex diseases such as metabolic syndrome and type 2 diabetes”—this work merely analyzed the compositions of gut microbiome among different groups (such as metabolic syndromes and diabetes). It barely addressed the microbiome’s “effects” on complex diseases. The authors did not reveal any causal relationships, nor any experimental mechanisms, therefore, this conclusion does not hold.

---

## Round 0.2 · Minor Revisions

· Academic Editor

Minor Revisions

Per email discussions, we are changing this to a Minor Revision in order to review the extensive language editing.

Reviewer 1 ·

Basic reporting

No additional comments, as the author has justified and applied the previous review's advice

Experimental design

No additional comments, as the author has justified and applied the previous review's advice

Validity of the findings

No additional comments, as the author has justified and applied the previous review's advice

Additional comments

The author has already made the necessary revisions based on my advice, so I have no further suggestions. However, the grammar could still be improved for better clarity. I suggest having it proofread by a native English speaker to enhance grammatical accuracy

---

## Round 0.3 · Minor Revisions

· Academic Editor

Minor Revisions

Review PDF and track changes word file are different. Please upload the matching files to avoid any confusion.

---

## Round 0.4 · accepted · Accept

· Academic Editor

Accept

The authors have addressed the reviewers' comments, and the paper is now in good shape.